# Towards Unsupervised Recognition of
# Token-level Semantic Differences in Related Documents

**Jannis Vamvas** and **Rico Sennrich**

Department of Computational Linguistics, University of Zurich

{vamvas,sennrich}@cl.uzh.ch

## Abstract

Automatically highlighting words that cause semantic differences between two documents could be useful for a wide range of applications. We formulate recognizing semantic differences (RSD) as a token-level regression task and study three unsupervised approaches that rely on a masked language model. To assess the approaches, we begin with basic English sentences and gradually move to more complex, cross-lingual document pairs. Our results show that an approach based on word alignment and sentence-level contrastive learning has a robust correlation to gold labels. However, all unsupervised approaches still leave a large margin of improvement. Code to reproduce our experiments is available.[1]

## 1 Introduction

A pair of documents can have semantic differences for a variety of reasons: For example, one document might be a revised version of the second one, or it might be a noisy translation. Highlighting words that contribute to a semantic difference is a challenging task (Figure 1). Previous work has studied word-level predictions in the context of interpretable textual similarity (Lopez-Gazpio et al., 2017) or evaluation of generated text (Fomicheva et al., 2022), but did not necessarily focus on semantic differences as the main target.

In this paper, we conceptualize the task of recognizing semantic differences (RSD) as a semantic *diff* operation. We assume that there are relatively few semantic differences and that many words are negative examples. Our goal is to label words using self-supervised encoders such as XLM-R (Conneau et al., 2020) without additional training data. Specifically, we investigate three simple metrics:

1. Performing word alignment and highlighting words that cannot be aligned;

Figure 1: While *diff* is a common tool for comparing code, highlighting the semantic differences in natural language documents can be more challenging. In this paper, we evaluate on synthetic documents that combine several such challenges, including cross-lingual comparison and non-monotonic sentence alignment.

2. Comparing document similarity with and without a word present in the document;
3. Comparing masked language modeling surprisal with and without the other document provided as context.

To evaluate these approaches automatically, we convert data from the SemEval-2016 Task for Interpretable Semantic Textual Similarity (iSTS; Agirre et al., 2016) into a token-level regression task, relabeling some words to better fit the goal of RSD.[2] We then programmatically create increasingly complex variations of this test set in order to study the robustness of the metrics: We add more negative examples, concatenate the sentences into synthetic documents, permute the order of sentences within the documents, and finally add a cross-lingual dimension by translating one side of the test set.

Our experiments show that the first metric correlates best to the gold labels, since measuring the alignability of words has a relatively consistent accuracy across complexity levels. However, while unsupervised approaches have the advantage of not requiring manual annotations, we find that there is a considerable gap to perfect accuracy, especially for cross-lingual document pairs. Future work could tackle the task by developing supervised models. Besides providing a baseline, unsupervised metrics could also serve as features for such models.

---

[1] https://github.com/ZurichNLP/
recognizing-semantic-differences

[2] In iSTS, opposites such as 'higher' and 'lower' are considered similar to each other, whereas we consider this to be a semantic difference worth highlighting.

## 2 Task Formulation

The goal of RSD is to analyze two word sequences $A = a_1, \ldots, a_n$ and $B = b_1, \ldots, b_m$ and to estimate, individually for each word, the degree to which the word causes a semantic difference between $A$ and $B$. For example, given the sentences *'Nice sweater!'* and *'Great news!'*, the correct labels would be close to 0 for *'Nice'* and *'Great'* and close to 1 for *'sweater'* and *'news'*.

Transformer-based encoders usually process documents as sequences of subword tokens. Our strategy is to predict labels for individual subwords and to average the labels of the subword tokens that make up a word. To make the notation more readable, we use $A$ and $B$ to refer to the tokenized sequences as well.

## 3 Recognition Approaches

**Alignability of a Word**  The final hidden states of a Transformer encoder (Vaswani et al., 2017) represent $A$ as a sequence of token embeddings $\mathbf{h}(A) = \mathbf{h}(a_1), \ldots, \mathbf{h}(a_n)$. In the same way, $B$ is independently encoded into $\mathbf{h}(B) = \mathbf{h}(b_1), \ldots, \mathbf{h}(b_m)$.

A simple approach to RSD is to calculate a soft token alignment between $\mathbf{h}(A)$ and $\mathbf{h}(B)$ and to identify tokens that are aligned with low confidence. A greedy alignment is usually calculated using the pairwise cosine similarity between hidden states (Jalili Sabet et al., 2020; Zhang et al., 2020). The prediction for a token $a_i$ is then given by:

$$\text{diff}_{\text{align}}(a_i) = 1 - \max_{b_j \in B} \cos(\mathbf{h}(a_i), \mathbf{h}(b_j)).$$

**Deletability of a Word**  Previous work has shown that encoders such as XLM-R may be fine-tuned such that the averages of their hidden states serve as useful sentence representations (Reimers and Gurevych, 2019; Gao et al., 2021). The similarity of two sentences can be estimated using the cosine similarity between these averages:

$$\text{sim}(A, B) = \cos(\text{avg}(A), \text{avg}(B))$$
$$= \cos(\frac{1}{|A|} \sum_{a_i \in A} \mathbf{h}(a_i), \frac{1}{|B|} \sum_{b_j \in B} \mathbf{h}(b_j)).$$

We approximate the similarity of a partial sequence $A \setminus a_i$, where $a_i$ is deleted, by excluding the token from the average:

$$\text{sim}(A \setminus a_i, B) = \cos(\text{avg}(A) - \frac{1}{|A|}\mathbf{h}(a_i), \text{avg}(B)).$$

The change in similarity when deleting $a_i$ can then serve as a prediction for $a_i$, which we normalize to the range $[0, 1]$:

$$\text{diff}_{\text{del}}(a_i) = \frac{\text{sim}(A \setminus a_i, B) - \text{sim}(A, B) + 1}{2}.$$

**Cross-entropy of a Word**  Encoders such as XLM-R or BERT (Devlin et al., 2019) have been trained using masked language modeling, which can be leveraged for our task. Let $H(a_i|A')$ be the cross-entropy under a masked language model that predicts the token $a_i$ given a context $A'$, where $a_i$ has been masked. By concatenating $B$ and $A'$ into an augmented context $BA'$, we can test whether the additional context helps the language model predict $a_i$. If the inclusion of $B$ does not reduce the cross-entropy, this could indicate that $B$ does not contain any information related to $a_i$:

$$\text{npmi}(a_i|A'; BA') = \frac{H(a_i|A') - H(a_i|BA')}{\max(H(a_i|A'), H(a_i|BA'))},$$

$$\text{diff}_{\text{mask}}(a_i) = 1 - \max(0, \text{npmi}(a_i|A'; BA')).$$

We base our score on normalized pointwise mutual information (npmi), with a simple transformation to turn it into $\text{diff}_{\text{mask}}$, a semantic difference score between 0 and 1.

## 4 Evaluation Design

### 4.1 Annotations

We build on annotated data from the SemEval-2016 Task 2 for Interpretable Semantic Textual Similarity (iSTS; Agirre et al., 2016). These data consist of English sentence pairs that are related but usually do not have the same meaning. The iSTS task is to group the tokens into chunks, to compute a chunk alignment and to label the chunk alignments with a type and a similarity score.

The iSTS annotations can be re-used for our task formulation by labeling each word with the inverse of the similarity score of the corresponding chunk alignment. If two chunks are aligned and have a high similarity, the words of the chunks receive a label close to 0. In contrast, if two aligned chunks have low similarity, or if a chunk is not aligned to any chunk in the other sentence, the words receive a label close to 1. Our evaluation metric is the Spearman correlation between the gold labels and the predicted labels across all words in the dataset.

Following iSTS, we do no consider punctuation, i.e., we exclude punctuation when calculating the

correlation. We deviate from the original iSTS annotations with regard to chunks with opposite meaning, marking them as differences. Further details are provided in Appendix A.

### 4.2 Negative Examples

Most sentence pairs in iSTS have a major semantic difference. To simulate a scenario with fewer such positive examples, we add additional negative examples to the test set. We use human-verified paraphrase pairs from PAWS (Zhang et al., 2019), where we label each word in the two sentences with a 0.

### 4.3 Synthetic Documents with Permutations

In addition, we experiment with concatenating batches of sentences into documents. The documents should be considered synthetic because the individual sentences are arbitrary and there is no document-level coherence. Despite this limitation, synthetic documents should allow us to test the approaches on longer sequences and even sequences where the information is presented in slightly different order.

Specifically, we keep document $A$ in place and randomly permute the order of the sentences in $B$ to receive $B^i$, where $i$ is the count of unordered sentences (*inversion number*). We control the degree of permutation via $i$, and sample a permuted document $B^i$ for each $B$ in the dataset.

### 4.4 Cross-lingual Examples

Finally, we evaluate on the recognition of differences between a document in English and a document in another language. For sentences from PAWS, we use existing human translations into German, Spanish, French, Japanese, Korean, and Chinese (PAWS-X; Yang et al., 2019). For iSTS, we machine-translate the sentences into these languages using DeepL, a commercial service. A risk of machine translation are accuracy errors that add (or eliminate) semantic differences. Our assumption is that any such errors are negligible compared to the absolute number of semantic differences in the dataset.[3]

In order to reduce annotation effort, we limit the evaluation to an English-centric setup where only the English sentence is annotated. When calculat-

---

[3]We manually analyzed a sample of 100 English–German translations. While five samples had issues with fluency, only one sample contained an accuracy error.

ing the evaluation metric, only the predictions on the English documents are considered.

## 5 Experimental Setup

We concatenate the 'headlines' and 'images' subsets of the iSTS dataset into a single dataset. We create a validation split by combining the iSTS training split with the PAWS-X validation split. Similarly, we create a test split by combining the iSTS test split with the PAWS-X test split. Appendix D reports data statistics.

We perform our experiments on the multilingual XLM-R model of size 'base'. In addition to the standard model, we also evaluate a model finetuned on SimCSE (Gao et al., 2021). SimCSE is a self-supervised contrastive learning objective that is commonly used for adapting a masked language model to the task of embedding sentences. To train XLM-R with SimCSE, we use 1M sentences from English Wikipedia and calculate sentence embeddings by averaging the final hidden states of the encoder. We train 10 checkpoints with different random seeds and report average metrics across the checkpoints. Details are provided in Appendix B.

## 6 Results

Table 1 presents validation results for the different approaches. We observe positive correlations throughout. Adding 50% paraphrases as negative examples to the test set leads to a decreased accuracy, indicating that imbalanced input is a challenge. When moving on to synthetic test documents composed of 5 sentences, the approaches tend to converge: word alignment becomes slightly less accurate, while diff$_{del}$ seems to benefit from the increased sequence length. Furthermore, when one of the two test documents is permuted with 5 inversions, recognizing the differences becomes slightly more difficult for most approaches.

Finally, cross-lingual document pairs clearly present a challenge to all approaches, since we observe a consistent decline in terms of the average correlation across the six language pairs. Appendix G provides results for the individual target languages, which show that comparing English documents to documents in Japanese, Korean or Chinese is particularly challenging. Appendices H and I juxtapose monolingual cross-lingual comparisons, illustrating that the latter are less accurate.

**Discussion of diff$_{align}$** When using XLM-R without further adaptation, the hidden states from the

| Approach | iSTS | + Negatives | + Documents | + Permuted | + Cross-lingual |
| --- | --- | --- | --- | --- | --- |
| | | 50% paraphrases | 5 sentences | 5 inversions | 6 language pairs |
| diff$_{align}$ | | | | | |
| – XLM-R (last layer) | 51.6 | 51.5 | 49.1 | 45.9 | 17.1 |
| – XLM-R (8th layer) | 56.9 | 51.0 | 49.5 | 48.1 | 28.7 |
| – XLM-R + SimCSE | **64.4** | **62.3** | **57.9** | **56.9** | **33.5** |
| diff$_{del}$ (XLM-R + SimCSE) | 29.6 | 9.8 | 29.3 | 25.8 | 4.0 |
| diff$_{mask}$ (XLM-R) | 51.2 | 46.1 | 49.4 | 49.7 | 24.9 |

Table 1: Comparison of different approaches and encoder models on the RSD validation split. The table reports word-level Spearman correlation to the gold labels. The variations are cumulative: the last column refers to a cross-lingual test set of permuted documents containing negative examples.

8th layer yield a more useful word alignment than the last layer, which confirms previous findings (Jalili Sabet et al., 2020; Zhang et al., 2020). Interestingly, we find that fine-tuning the model with SimCSE strongly improves these results. Even though SimCSE is an unsupervised sentence-level objective, the results suggest that learning sentence-level representations also improves the quality of word alignment. This is in line with a related finding of Leiter (2021) that supervised fine-tuning on NLI can improve the explainability of BERTScore.

**Discussion of diff$_{del}$** Relying on the deletability of a word has a lower accuracy than word alignment. In Appendix F we test more complex formulations of diff$_{del}$ and find that accuracy can be improved by deleting bigrams and trigrams in addition to subword unigrams, but does not reach the accuracy of diff$_{align}$.

**Discussion of diff$_{mask}$** Using the cross-entropy of masked language modeling yields some competitive results on longer documents. However, latency measurements show that this approach is much slower than the other approaches, since the documents need to be re-encoded for each masked word (Appendix C).

For the best-performing approach, diff$_{align}$ with SimCSE, we report results on the test split in Appendix E. The test results confirm the patterns we have observed on the validation set.

## 7 Related Work

The idea of highlighting semantic differences or similarities on the level of individual words has influenced several research areas, notably interpretable semantic textual similarity (Lopez-Gazpio et al., 2017) and the evaluation of generated text (Freitag et al., 2021; Fomicheva et al., 2022; Zerva et al., 2022; Rei et al., 2023). Other application areas include content synchronization across documents (Mehdad et al., 2012) and the detection of relevant differences in legal texts (Li et al., 2022).

Some previous work has explored unsupervised or indirectly supervised approaches to such tasks. Leiter (2021) measured the alignability of words to identify words that negatively impact the quality of translations. Word alignment has also been used to estimate sentence similarity (Mathur et al., 2019; Zhang et al., 2020), and Lee et al. (2022) fine-tuned such a similarity metric with sentence-level supervision in order to promote word-level interpretability.

Deleting words to measure their effect on sentence similarity is related to occlusion-based feature attribution methods (Robnik-Šikonja and Kononenko, 2008). Yao et al. (2023) used a similar method to evaluate sentence representations. Finally, the effect of context on cross-entropy (*cross-mutual information*) has previously been analyzed in the context of machine translation (Bugliarello et al., 2020; Fernandes et al., 2021).

## 8 Conclusion

We formulated the task of recognizing semantic differences (RSD) between two documents as a token-level regression task and analyzed several unsupervised approaches towards this task. Our experiments use annotations from iSTS, which we programmatically recombined into more challenging variants of the test set. We found that the alignability of a word is the most accurate measure,

especially when the word alignment is computed with a SimCSE-adapted masked language model.

## Limitations

Like many NLP tasks, RSD is difficult to formalize. In some edge cases, it is unclear which words 'cause' a semantic difference, given that natural language is not entirely compositional. For example, it is unclear which specific words should be highlighted in the pair 'flights from New York to Florida' and 'flights from Florida to New York'. Since iSTS focuses on the annotation of syntactic chunks, we follow that convention and assign the same label to all words in a chunk.

Another challenge is distinguishing between semantic and non-semantic differences. This paper re-uses annotations from the iSTS datasets and thus inherits its guidelines (except for phrases with opposite meaning, where we stress semantic difference, while iSTS stresses semantic relatedness).

Furthermore, we assume that semantics is invariant to machine translation (MT) into another language. In practice, MT might introduce errors that add or eliminate semantic differences, and human translation might be more reliable. However, we expect there to be little correlation between the gold differences and any accuracy errors that might be introduced by MT. Moreover, there are no low-resource language pairs in the experiment, where the risk of MT accuracy errors would be higher.

A methodical limitation is that our experiments are based on synthetic documents that we compiled programmatically from human-annotated sentences, such as headlines and image captions. Our assumption is that synthetic documents can help us learn about the accuracy of different recognition approaches and that the findings will roughly translate to natural documents. Finally, we assume that the gold labels that human annotators originally applied to individual sentence pairs remain valid when the sentence pairs are embedded in an arbitrary context.

## Acknowledgements

This work was funded by the Swiss National Science Foundation (project MUTAMUR; no. 176727). We would like to thank Chantal Amrhein and Marek Kostrzewa for helpful feedback.

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

## A    Converting iSTS into an RSD Dataset

We derive our test set from the 2016 iSTS dataset described by Agirre et al. (2016). The original URLs of the data are:

- Train: `http://alt.qcri.org/semeval2016/task2/data/uploads/train_2015_10_22.utf-8.tar.gz`

- Test: `http://alt.qcri.org/semeval2016/task2/data/uploads/test_goldstandard.tar.gz`

Examples from the iSTS dataset consist of two tokenized sentences and the corresponding chunk alignments. Every chunk alignment is annotated with an alignment type and a similarity score between 1 and 5, where 5 means semantic equivalence. If no corresponding chunk is found in the other sentence, the chunk has a similarity score of NIL, which corresponds to 0.

We determine word-level labels for our difference recognition task as follows:

1. The similarity score for a chunk is applied to the individual words in the chunk.

2. The label is calculated as $1 - \text{score}/5$.

A small number of chunks are aligned with the OPPO type, which denotes opposite meaning. The iSTS annotation guidelines encouraged the annotators to assign relatively high similarity scores to strong opposites. However, in the context of RSD, we regard opposite meaning as a difference. To account for this, we re-label the OPPO alignments with the similarity score 0. An example are the words 'lower' and 'higher' in Appendix H, which were originally aligned with a score of 4.

When creating cross-lingual examples, we translate all the sentence pairs $(A, B)$ in the dataset from English to the target languages, resulting in the translations $A'$ and $B'$. We then combine them into two cross-lingual examples: $A$ compared to $B'$ and $B$ compared to $A'$. We only consider the predictions on $A$ and $B$, respectively, when calculating the correlations, since we lack gold labels for the translations.

The converted data are available for download at `https://huggingface.co/datasets/ZurichNLP/rsd-ists-2016`.

## B    SimCSE Training Details

Positive examples for SimCSE (Gao et al., 2021) are created by applying two random dropout masks to a sentence $S_i$, which results in two encodings $\mathbf{h}(S_i)$ and $\mathbf{h}'(S_i)$. Negative examples are arbitrary sentence pairs in the mini-batch. The training objective is:

$$\ell_i = -\log \frac{e^{\text{sim}(\mathbf{h}(S_i),\mathbf{h}'(S_i))/\tau}}{\sum_{j=1}^{N} e^{\text{sim}(\mathbf{h}(S_i),\mathbf{h}'(S_j))/\tau}},$$

where $N$ is the batch size, $\tau$ is a temperature parameter, and $\text{sim}(\mathbf{h}, \mathbf{h}')$ is a similarity measure. In our experiments, we use the cosine similarity of the average hidden states as a similarity measure:

$$\text{sim}_{\text{avg}}(\mathbf{h}, \mathbf{h}') = \cos(\text{avg}(\mathbf{h}), \text{avg}(\mathbf{h}')).$$

For training XLM-R on SimCSE, we use a maximum sequence length of 128. Otherwise we use the hyperparameters recommended by Gao et al. (2021), namely a batch size of 512, a learning rate of 1e-5, and $\tau = 0.05$, and we train the model for one epoch.

A model checkpoint is made available at the URL: `https://huggingface.co/ZurichNLP/unsup-simcse-xlm-roberta-base`.

## C    Latency Measurements

| Approach | Time per 1000 Tokens |
|---|---:|
| diff$_{\text{align}}$ | 0.44 s |
| diff$_{\text{del}}$ | 0.45 s |
| diff$_{\text{mask}}$ | 97.04 s |

Table 2: Comparison of inference time. We use XLM-R of size 'base' with batch size of 16 on an RTX 3090 GPU. We report the number of seconds needed to make predictions for 1000 tokens of the iSTS test set.

# D  Dataset Statistics

| Dataset | Document pairs | Tokens | Labels $< 0.5$ | Labels $\geq 0.5$ | Unlabeled Tokens |
|---|---|---|---|---|---|
| *Validation split* | | | | | |
| iSTS | 1506 | 27046 | 64.5% | 28.2% | 7.3% |
| + Negatives (50% paraphrases) | 3012 | 92443 | 80.6% | 8.2% | 11.1% |
| + Documents (5 sentences) | 602 | 92389 | 80.7% | 8.2% | 11.1% |
| + Permuted (5 inversions) | 602 | 92389 | 80.7% | 8.2% | 11.1% |
| + Cross-lingual (DE) | 1204 | 183609 | 40.6% | 4.1% | 55.2% |
| *Test split* | | | | | |
| iSTS | 750 | 13801 | 70.0% | 22.9% | 7.0% |
| + Negatives (50% paraphrases) | 1500 | 46671 | 82.4% | 6.8% | 10.8% |
| + Documents (5 sentences) | 300 | 46671 | 82.4% | 6.8% | 10.8% |
| + Permuted (5 inversions) | 300 | 46671 | 82.4% | 6.8% | 10.8% |
| + Cross-lingual (DE) | 600 | 92649 | 41.3% | 3.4% | 55.3% |

Table 3: Statistics for the validation and test sets. The variations are cumulative, e.g., the bottom row combines all previous variations.

# E  Test Results

| Approach | iSTS | + Negatives | + Documents | + Permuted | + Cross-lingual |
|---|---|---|---|---|---|
| $\text{diff}_{\text{align}}$ XLM-R + SimCSE | 62.1 | 61.1 | 57.0 | 55.1 | 31.7 |

Table 4: Results on the RSD test split for the best-performing approach. The table reports word-level Spearman correlation to the gold labels.

# F  Ablations for $\text{diff}_{\text{del}}$

| Approach | iSTS | + Negatives | + Documents | + Permuted | + Cross-lingual |
|---|---|---|---|---|---|
| $\text{diff}_{\text{del}}$ XLM-R + SimCSE | 29.6 | 9.8 | 29.3 | 25.8 | 4.0 |
| – unigrams and bigrams | 35.2 | 10.5 | 32.3 | 28.3 | 4.3 |
| – unigrams, bigrams and trigrams | 38.1 | 10.2 | 33.5 | 29.2 | 4.4 |
| – unigrams with re-encoding | 42.4 | 11.5 | 24.0 | 22.2 | 6.2 |

Table 5: Evaluation of more complex variants of $\text{diff}_{\text{del}}$ on the validation split. Measuring the deletability of bigrams or trigrams of subword tokens (instead of only single tokens) tends to improve Spearman correlation. In contrast, encoding the partial sentences from scratch (instead of encoding the full sentence once and then excluding hidden states from the mean) does not consistently improve the metric.

# G Additional Results

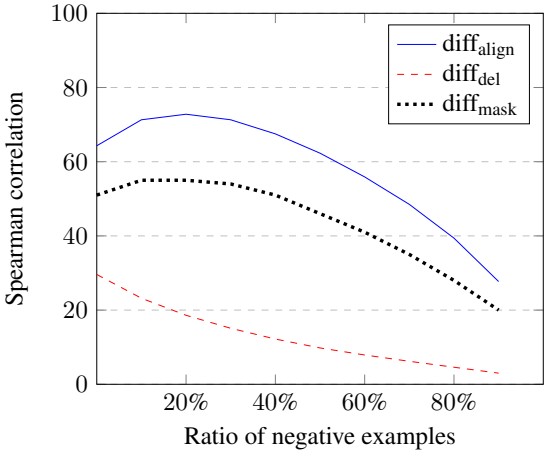

(a) Recognizing differences among an increasing amount of negative examples, which do not have semantic differences.

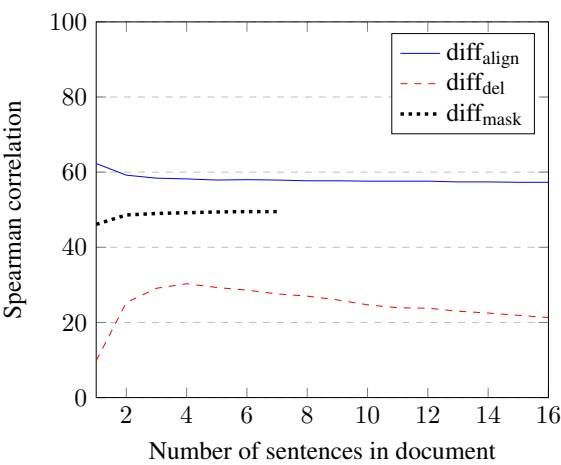

(b) Recognizing differences between increasingly longer synthetic documents (ratio of negative sentence pairs: 50%). $\text{diff}_{\text{mask}}$ is limited to the maximum sequence length of the masked language model.

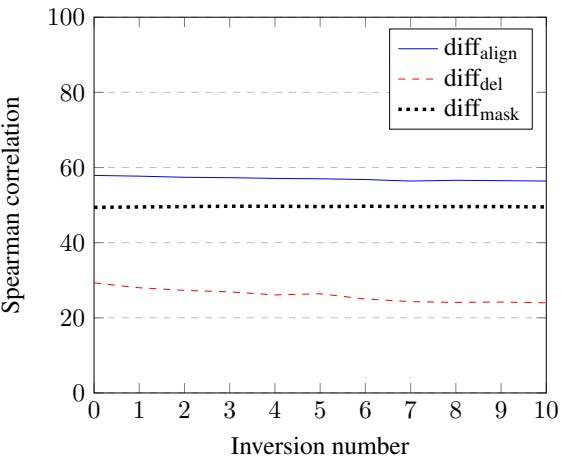

(c) Recognizing differences between increasingly permuted documents (document length: 5 sentences; ratio of negative sentence pairs: 50%).

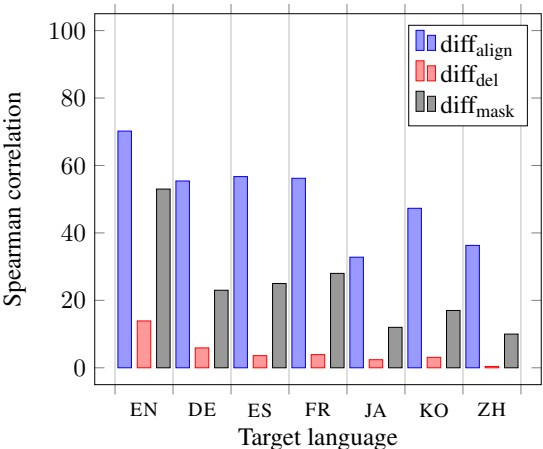

(d) Recognizing differences between English source sentences and target sentences in various languages (ratio of negative examples: 50%).

Figure 2: Additional results for variants of the validation set. The $\text{diff}_{\text{align}}$ and $\text{diff}_{\text{del}}$ approaches use an XLM-R model fine-tuned with SimCSE; the $\text{diff}_{\text{mask}}$ approach uses the standard XLM-R model.

## H Examples (English)

**Gold labels**

Chinese shares close higher Friday. Syria opposition threatens to quit talks.
*Syria opposition agrees to talks. Chinese shares close lower Wednesday.*

**diff_align XLM-R (last layer)**

Chinese shares close higher Friday. Syria opposition threatens to quit talks.
*Syria opposition agrees to talks. Chinese shares close lower Wednesday.*

**diff_align XLM-R + SimCSE**

Chinese shares close higher Friday. Syria opposition threatens to quit talks.
*Syria opposition agrees to talks. Chinese shares close lower Wednesday.*

**diff_del XLM-R + SimCSE**

Chinese shares close higher Friday. Syria opposition threatens to quit talks.
*Syria opposition agrees to talks. Chinese shares close lower Wednesday.*

**diff_mask XLM-R**

Chinese shares close higher Friday. Syria opposition threatens to quit talks.
*Syria opposition agrees to talks. Chinese shares close lower Wednesday.*

Figure 3: Predictions for an example document pair with two English sentences each. The example contains one inversion, since the two sentences in the second document have been swapped. The gold labels are derived from the SemEval-2016 Task 2 for Interpretable Semantic Textual Similarity (Agirre et al., 2016).

## I Examples (Cross-lingual)

**Gold labels**

Chinese shares close higher Friday. Syria opposition threatens to quit talks.
*L'opposition syrienne accepte des pourparlers. Les actions chinoises clôturent en baisse mercredi.*

**diff_align XLM-R (last layer)**

Chinese shares close higher Friday. Syria opposition threatens to quit talks.
*L'opposition syrienne accepte des pourparlers. Les actions chinoises clôturent en baisse mercredi.*

**diff_align XLM-R + SimCSE**

Chinese shares close higher Friday. Syria opposition threatens to quit talks.
*L'opposition syrienne accepte des pourparlers. Les actions chinoises clôturent en baisse mercredi.*

**diff_del XLM-R + SimCSE**

Chinese shares close higher Friday. Syria opposition threatens to quit talks.
*L'opposition syrienne accepte des pourparlers. Les actions chinoises clôturent en baisse mercredi.*

**diff_mask XLM-R**

Chinese shares close higher Friday. Syria opposition threatens to quit talks.
*L'opposition syrienne accepte des pourparlers. Les actions chinoises clôturent en baisse mercredi.*

Figure 4: Predictions for the example in Figure 1, where the second document has been machine-translated into French. Note that the non-English documents in our test examples do not have gold labels. In our experiments, we only evaluate the predictions for the English document.