# OpenReview forum: "Towards Unsupervised Recognition of Token-level Semantic Differences in Related Documents"
_EMNLP/2023/Conference — EMNLP 2023 Main_

### Official Review · Reviewer_RPjL · 2023-08-03

**Soundness:** 3

**Excitement:**

3: Ambivalent: It has merits (e.g., it reports state-of-the-art results, the idea is nice), but there are key weaknesses (e.g., it describes incremental work), and it can significantly benefit from another round of revision. However, I won't object to accepting it if my co-reviewers champion it.

**Paper Topic And Main Contributions:**

This paper discusses a novel task, recognizing semantic differences (RSD), which aims to recognize token-level semantic differences between two documents. The authors propose three unsupervised methods for RSD,  Alignability of a Word, Deletability of a Word, and Cross-entropy of a Word, and then evaluate these methods through experiments using a dataset created from the iSTS task dataset.

**Reasons To Accept:**

- This paper describes the task definition of RSD, which may be novel compared with previous works.
- Each method is clearly explained.
- The authors create an evaluation dataset specific to RSD, including several settings (negative examples, permutations, and cross-lingual examples), and use it to evaluate the proposed methods.

**Reasons To Reject:**

- The reviewer can not judge the effectiveness of the proposed methods because there are no baselines in the experiments.
- The Authors should clearly state their reasons for focusing only on the unsupervised approach. If evaluation data can be created automatically, training data can also be created on the same principle, so the reviewer thinks the supervised methods can also be tried.
- "Alignability of a Word" is a reasonable approach, but there is a concern that the technical novelty may be slightly weak.

**Reproducibility:**

4: Could mostly reproduce the results, but there may be some variation because of sample variance or minor variations in their interpretation of the protocol or method.

**Reviewer Confidence:**

3: Pretty sure, but there's a chance I missed something. Although I have a good feel for this area in general, I did not carefully check the paper's details, e.g., the math, experimental design, or novelty.

---

> ### Author Rebuttal · Authors · 2023-08-24
>
> Thank you for the review. Below we respond to your concerns:
>
> > _"The reviewer can not judge the effectiveness of the proposed methods because there are no baselines in the experiments."_
>
> Yes! Because we are investigating a novel task, there are no baselines from previous work that we could readily use. However, we strongly believe that novel tasks should have a place at EMNLP, and punishing papers for lack of external baselines would make this type of contribution hard to publish. We report results for three approaches, which can serve as baselines for future work.
>
> > _"If evaluation data can be created automatically, training data can also be created on the same principle, so the reviewer thinks the supervised methods can also be tried."_
>
> The evaluation data we used are not created fully automatically but depend on human annotations from iSTS and PAWS (word-level semantic similarity / sentence-level paraphrasticity). It would be very costly to annotate a sizeable training set using the same approach.
>
> > _"The Authors should clearly state their reasons for focusing only on the unsupervised approach."_
>
> Thanks for this recommendation. We will follow your recommendation and will discuss more explicitly in the text why our short paper focuses on unsupervised approaches – especially the cost of manually annotating supervised training data and possible future work in this direction.
>
> Thanks again for your review, and please consider adjusting your Soundness/Excitement scores if this author response has addressed your concerns.

---

### Official Review · Reviewer_4d51 · 2023-08-04

**Soundness:** 4

**Excitement:**

3: Ambivalent: It has merits (e.g., it reports state-of-the-art results, the idea is nice), but there are key weaknesses (e.g., it describes incremental work), and it can significantly benefit from another round of revision. However, I won't object to accepting it if my co-reviewers champion it.

**Paper Topic And Main Contributions:**

This paper proposes an approach aiming at identifying words introducing semantic differences, by studying the differences between pairs of documents. The method is based on  word alignment and sentence-level contrasts. unsupervised approaches are analyzed. Different metrics are proposed: word alignment (highlighting words that cannot be aligned), document similarity (with and without the studied word), masked language modeling surprisal (with and without the other document as context). A test set is created, introducing different levels of complexity. Alignability, deletability and cross-entropy of words are defined and compared. The word alignability provides the most accurate measure.

**Reasons To Accept:**

This sort paper is clear and well presented. The proposal is simple, and results potentially interesting in different perspectives, including in  approaches focusing on the evaluation of information transfer.

**Reasons To Reject:**

Not really exciting, no great originality

**Reproducibility:**

4: Could mostly reproduce the results, but there may be some variation because of sample variance or minor variations in their interpretation of the protocol or method.

**Reviewer Confidence:**

4: Quite sure. I tried to check the important points carefully. It's unlikely, though conceivable, that I missed something that should affect my ratings.

---

### Official Review · Reviewer_uPJb · 2023-08-04

**Typos Grammar Style And Presentation Improvements:** l325
**Soundness:** 4

**Excitement:**

3: Ambivalent: It has merits (e.g., it reports state-of-the-art results, the idea is nice), but there are key weaknesses (e.g., it describes incremental work), and it can significantly benefit from another round of revision. However, I won't object to accepting it if my co-reviewers champion it.

**Paper Topic And Main Contributions:**

The paper attempts to find semantic differences in documents using unsupervised techniques. Three different approaches are tried, all based on masked language models. The first looks at token alignment, the second considers deletion and the third on normalized pointwise mutual information. The evaluation corpus is constructed using a combination of iSTS and PAWS. For the cross-lingual component, the iSTS data is automatically translated into the foreign language. The authors find that the best results are found with the alignment based approach.

**Reasons To Accept:**

The paper includes a nice approach to creating a dataset of similar and dissimilar pairs.

As an unsupervised approach, the main idea of the paper can be further integrated into a supervised system (as also suggested by the authors), offering scope for further improving the detection of semantic differences.

The paper is well written with the appendices used to supply additional information such as the technical aspects of the work and additional results.

**Reasons To Reject:**

The authors use a commercial machine translation system to translate dissimilar pairs and assume that it makes negligible errors. Without verification, this may be a reason for the low cross-lingual results.


**Reproducibility:**

4: Could mostly reproduce the results, but there may be some variation because of sample variance or minor variations in their interpretation of the protocol or method.

**Reviewer Confidence:**

2: Willing to defend my evaluation, but it is fairly likely that I missed some details, didn't understand some central points, or can't be sure about the novelty of the work.

---

> ### Author Rebuttal · Authors · 2023-08-24
>
> Thank you for the positive review. Below we respond to your concern about MT:
>
> > _"The authors use a commercial machine translation system to translate dissimilar pairs and assume that it makes negligible errors. Without verification, this may be a reason for the low cross-lingual results."_
>
> There are several considerations that give us confidence that MT is acceptable:
>
> * We manually checked a sample of 100 EN–DE translations. While five samples had issues with fluency, only one sample contained an accuracy error.
>
> * There are no low-resource language pairs in the experiment. It is possible that one of the other language pairs has a higher rate of accuracy errors than the one we checked, but we do not expect it to be an order of magnitude higher.
>
> * The difference phenomena in the test set tend to be clear-cut (e.g., "Friday" vs. "Wednesday" or "higher share prices" vs. "lower share prices"). They are derived from the combination of thematically related but ultimately arbitrary documents, and not from translation errors. Therefore, we expect there to be little correlation between the gold differences and any MT accuracy errors introduced by our evaluation approach.
>
> We will add the points outlined above to the Limitations section to make our reasoning more explicit.
>
> Thanks again for the positive review, and please consider adjusting your Soundness/Excitement scores if our response addresses your concern about the use of MT.

---

### Meta-Review · Area_Chair_4Tnz · 2023-09-18

**Recommendation:** 4

**Metareview:**

All the reviewers agree that the paper presents an interesting method and that it is well written. The reviewers found some of the ideas novel and did not have many criticisms.

One of the reviewers questioned the use of a commercial machine translation, but without evaluating its performance. In the rebuttal the authors carried out a small scale evaluation showing that the errors introduced by MT are minimal and likely to have limited impact on the results.

The lack of comparison with baselines was  criticised by one reviewer because without them it is difficult to know how good the methods are. The lack of baselines is justified by the authors by the novelty of the task. However, they could have tried at least some naive baselines.

Overall, the reviews indicate that this is a strong paper which contains some novel ideas especially given that it is a short paper which cannot pack too much information.

---

### Decision · Program_Chairs · 2023-10-07

**Decision:**

Accept-Main

**Comment:**

All the reviewers agree that the paper presents an interesting method and that it is well written. The reviewers found some of the ideas novel and did not have many criticisms.

One of the reviewers questioned the use of a commercial machine translation, but without evaluating its performance. In the rebuttal the authors carried out a small scale evaluation showing that the errors introduced by MT are minimal and likely to have limited impact on the results.

The lack of comparison with baselines was  criticised by one reviewer because without them it is difficult to know how good the methods are. The lack of baselines is justified by the authors by the novelty of the task. However, they could have tried at least some naive baselines.

Overall, the reviews indicate that this is a strong paper which contains some novel ideas especially given that it is a short paper which cannot pack too much information.